# Evaluation of Specific Cellular and Humoral Immune Response to *Toxoplasma gondii* in Patients with Autoimmune Rheumatic Diseases Immunomodulated Due to the Use of TNF Blockers

**DOI:** 10.3390/biomedicines11030930

**Published:** 2023-03-17

**Authors:** Cristhianne Molinero Ratkevicius Andrade, Aline Caroline de Lima Marques, Rodolfo Pessato Timóteo, Ana Carolina de Morais Oliveira-Scussel, Fernanda Bernadelli De Vito, Marcos Vinícius da Silva, José Roberto Mineo, Reginaldo Botelho Teodoro, Denise Bertulucci Rocha Rodrigues, Virmondes Rodrigues Júnior

**Affiliations:** 1Laboratory of Immunology, Triângulo Mineiro Federal University, Uberaba 38025-350, Brazil; 2Laboratory of Hematological Research, Triângulo Mineiro Federal University, Uberaba 38025-350, Brazil; 3Laboratory of Parasitology, Triângulo Mineiro Federal University, Uberaba 38025-350, Brazil; 4Institute of Biomedical Sciences, Laboratory of Immunoparasitology, Federal University of Uberlândia, Uberlândia 38408-100, Brazil; 5Department of Clinical Medicine, Triângulo Mineiro Federal University, Uberaba 38025-180, Brazil; 6Laboratory of Biopathology and Molecular Biology, University of Uberaba, Uberaba 38055-500, Brazil; 7INCT-Neuroimmune Modulation, Uberaba 38025-350, Brazil

**Keywords:** TNF-α blockers, immunoglobulins, *Toxoplasma gondii*, immunosuppression, toxoplasmosis, immune response

## Abstract

(1) Background: TNF antagonists have been used to treat autoimmune diseases (AD). However, during the chronic phase of toxoplasmosis, TNF-α and TNFR play a significant role in maintaining disease resistance and latency. Several studies have demonstrated the risk of latent infections’ reactivation in patients infected with toxoplasmosis. Our objective was to verify whether patients with autoimmune rheumatic diseases, who use TNF antagonists and/or synthetic drugs and had previous contact with *Toxoplasma gondii* (IgG^+^), present any indication of an increased risk of toxoplasmosis reactivation. (2) Methods: Blood samples were collected, and peripheral blood mononuclear cells (PBMCs) were evaluated after stimulation with antigens of *Toxoplasma gondii*, with anti-CD3/anti-CD28 or without stimulus, at 48 and 96 h. CD69^+^, CD28^+^, and PD-1 stains were evaluated, in addition to intracellular expression of IFN-γ, IL-17, and IL-10 by CD4^+^ and the presence of regulatory CD4^+^ T cells by labeling CD25^+^, FOXP3, and LAP. The cytokines IL-2, IL-4, IL-6, IL-10, IFN-γ, TNF-α, and IL-17 were measured in the culture supernatant after 96 h. Serology for IgG and IgG1 was evaluated. (3) Results: There were no differences in the levels of IgG and IgG1 between the groups, but the IgG1 avidity was reduced in the immunobiological group compared to the control group. All groups exhibited a significant correlation between IgG and IgG1 positivity. CD4^+^ T lymphocytes expressing PD-1 were increased in individuals suffering from autoimmune rheumatic diseases and using disease-modifying antirheumatic drugs. In addition, treatment with TNF blockers did not seem to influence the populations of regulatory T cells and did not interfere with the expression of the cytokines IFN-γ, IL-17, and IL-10 by CD4^+^ cells or the production of cytokines by PBMCs from patients with AD. (4) Conclusions: This study presents evidence that the use of TNF-α blockers did not promote an immunological imbalance to the extent of impairing the anti-*Toxoplasma gondii* immune response and predisposing to toxoplasmosis reactivation.

## 1. Introduction

Toxoplasmosis is one of the most prevalent zoonoses in Brazil and worldwide, with a varied distribution according to the population studied. Although it is a self-limited disease in immunocompetent subjects, it can have serious consequences in immunosuppressed patients [1,2]. The immune response is crucial for maintaining toxoplasmosis latency. For this reason, there is great concern about the prevention and monitoring of acute toxoplasmosis in immunocompromised patients, such as transplanted or HIV-positive patients, pregnant women, and newborns, who are at increased risk of developing severe diseases [3,4,5].

Interferon-gamma (IFN-γ) and its receptor, IFN-γR, play a major role in containing the *Toxoplasma gondii* infection during the acute phase. However, during the chronic phase, the action of this cytokine alone is not sufficient. The tumor necrosis factor alpha (TNF-α) and TNFR are therefore required to maintain disease resistance and latency in hematopoietic cells and other host cells [6].

In recent years, TNF antagonists have been used to treat autoimmune diseases and have provided, in many cases, better control and quality of life for related patients [7]. However, several studies have shown some risks associated with the use of these medications [8,9,10,11], such as the appearance of opportunistic infections [12,13]. Some diseases, such as tuberculosis, are closely monitored due to their high risk of reactivation in these patients [12,14]. With regard to this disease, the use of anti-TNF-α in vitro promotes a negative modulation of cytokines that are essential for the development of the Th1 [15].

In toxoplasmosis, the treatment with the TNF-α antagonist, etanercept, in mice infected with *T. gondii* promotes a significant decrease in TNF-α levels which are associated with a higher number of brain cysts [16]. In addition, there are other experiments that demonstrated the risk of reactivation in mice [16,17] and some case reports of patients who developed manifestations of toxoplasmic infection [18,19].

These data suggest that individuals using TNF blockers may be at risk of toxoplasmosis reactivation [16,17,18,19]. However, in Brazil, there is still no screening for toxoplasmosis before the administration of TNF-α blockers or during monitoring throughout the treatment [20,21]. Thus, the fact that there are reports on the reactivation of this disease, associated with the scarcity of data on the maintenance of latent toxoplasmosis before and during therapy with immunobiologicals, reinforces the need for further studies that evaluate the immune response against *T. gondii* in these patients.

Since the inhibition of TNF-α may predispose to failure to maintain the chronicity of the disease, with the consequent conversion of bradyzoites into tachyzoites due to the impairment of anti-*T. gondii* immune response [16,22], our study evaluated cellular markers and the production of important cytokines for controlling the parasite in patients with rheumatic diseases using TNF antagonists.

## 2. Materials and Methods

### 2.1. Study Design

Patients treated at the rheumatology service of the UFTM General Hospital (Uberaba, State of Minas Gerais, Brazil) were invited to participate in the study. Individuals with positive serology for *T. gondii* and without autoimmune diseases, who were treated in the other sectors of the hospital, were invited to join the control group (CG). Patients aged between 18 and 60 years old and who had IgG anti-*T. gondii* were selected, and their blood was collected for cell culture. They had to be taking their medication properly and should not have presented any diagnosis of opportunistic disease or any other acute condition, such as flu, recent heart attack, and bone fracture, among others that could interfere in the analysis of the in vitro response against *Toxoplasma gondii*.

Additionally, using the Management Application for University Hospitals (AGHU), an online medical record survey was conducted to verify the following data: diagnosed autoimmune disease, date of diagnosis, treatment time, and type of medication used.

The research participants were grouped according to the type of medication they were using, namely synthetic drugs or biological immunomodulators. The synthetic drugs include medications that control symptoms, such as analgesics, non-steroidal anti-inflammatory drugs, and corticosteroids (they have immunosuppressive action when used for a long time). Disease-modifying antirheumatic drugs (DMARDs) were also included [23], such as sulfasalazine, hydroxychloroquine sulfate, leflunomide, and methotrexate (MTX). Biological immunomodulators are considered biological DMARDs and include infliximab, etanercept, certolizumab pegol, golimumab, and adalimumab, all blockers of the TNF-α cytokine [24], which was the focus of the present study. Figure 1 presents the groups of patients and the criteria for their inclusion in each group.

All patients were informed about the purpose of the study, and only those who accepted and signed the free and informed consent form (ICF) were included. This project was previously approved by the Research Ethics Committee (CEP) of the Federal University of Triângulo Mineiro (UFTM), under the protocol number 1,870,741, and complies with the Declaration of Helsinki.

### 2.2. Maintenance of T. gondii Cultures and Obtaining Parasites

The tachyzoite forms of the *T. gondii* RH strain were maintained in cell culture using HeLa cell lines [25]. These cells were infected with *T. gondii* tachyzoites that were maintained through serial passages in Roswell Park Memorial Institute (RPMI, Gibco Thermo Fisher Scientific, Waltham, MA, USA) medium with 5% fetal bovine serum every 48–72 h. Free parasites were collected with a cell scraper and partially purified by forced passage through a 13 × 4 mm needle and rapid centrifugation (70× *g* for two minutes at 4 °C), which was performed to remove cell debris. The supernatant was collected and washed twice (900× *g* for 10 min at 4 °C) with 0.01 M phosphate buffered saline (PBS, pH 7.2). The final pellet of the parasite suspension was resuspended in PBS and stored at −80 °C until the soluble *T. gondii* antigen was prepared.

### 2.3. Preparation of Soluble Toxoplasma gondii Antigens (STAg) for Immunoenzymatic Assays and for Cell Culture

Parasitic suspensions containing approximately 10^8^ tachyzoites/mL were resuspended in an ultrapure water containing the cocktail of COMPLETE^TM^ protease inhibitors (COMPLETE^TM^ ULTRA Tablets, Mini, EASYpack, Roche Applied Science, Switzerland) and subjected to 10 rapid cycles of freezing in liquid nitrogen and heating in a water bath at 37 °C. Then, osmolarity was adjusted with 10× sterile PBS, and eight cycles of ultrasound at 60 Hz in an ice bath were conducted for five minutes. After centrifugation at 10,000× *g* for 30 min at 4 °C, the supernatant was collected and aliquoted. For use in cell culture, the same protocol was followed but with the exclusion of the COMPLETE^TM^ protease inhibitor cocktail and the ultrasound cycles and with the addition of a 0.22 μm pore membrane antigen filtration step (Millipore, USA). Crude antigens were aliquoted into sterile tubes. Protein concentration was determined [26] in both crude soluble antigens, and aliquots were stored in a freezer at −80 °C until use.

### 2.4. ELISA Test for the Detection of Anti-Toxoplasma gondii Total IgG, Its Subclass IgG1, and ELISA-Avidity for Total IgG and IgG1 Anti-T. gondii

The tests for the detection of anti-*Toxoplasma gondii* antibodies, total IgG, and its subclass, IgG1, in the patients’ serum samples were performed using the enzyme-linked immunosorbent assay (ELISA). The high-affinity plaques (Thermo Scientific^TM^ Nunc^TM^, Waltham, Massachusetts, USA) were sensitized with STAg (10 μg/mL) diluted in 0.06 M carbonate-bicarbonate buffer (pH 9.6) and incubated for 18 h at 4 °C. Subsequently, the plaque for IgG test was washed three times with PBS containing 0.05% Tween 20 (PBS-T) and blocked with PBS-T containing 5% skimmed milk powder (Molico, Nestle, São Paulo, SP, Brazil, PBS -T-M5%) for one hour at room temperature. After being washed again, the serum samples were diluted 1:64 in PBS-T-M5% and incubated for one hour at 37 °C. After six washes, anti-human IgG antibody (1:2000) conjugated to peroxidase (IgG/HRP, DAKO) was added and incubated for one hour at 37 °C. Following another washing, the reaction was revealed by the addition of the enzymatic substrate 1,2 orthophenylenediamine (OPD, Dako) with 3% of H_2_O_2_ (30%) diluted in ultrapure water.

For the evaluation of the subclasses, the same processes were followed for the sensitizing plaques and washes. Blocking was performed with PBS-T containing 1% bovine serum albumin (1% PBS-T-BSA) for one hour at room temperature. After the washes, the serum samples were also diluted to the ratio 1:64 in 1% PBS-T-BSA and incubated for two hours at 37 °C. After six washes, the anti-human IgG1 (1:1000, BD Pharmigen^TM^), conjugated with biotin, was incubated for one hour at 37 °C. The plates were then washed six times, and streptavidin conjugated to peroxidase was added (1:1000) and incubated for 30 min in the dark at room temperature. After another washing, the reaction was revealed.

The absorbance values were determined on a microtiter plate reader at 490 nm. Both the positive and negative controls were included on the plate. The levels of antibodies were expressed in the ELISA index (EI) according to the formula EI = optical density (OD) sample/cut off, where the cut off was calculated as the mean OD of the negative control sera plus three standards deviations. EI values > 1.2 were considered positive for total IgG and IgG1, while borderline reactivity values close to EI = 1.0 were not considered positive.

A test was also performed to evaluate the avidity of total IgG and subclass IgG1. The same ELISA procedure was performed for the positive serum samples, but there was an additional step after incubation with samples from the patients where 8 M urea solution diluted in PBS 1× was placed in one of the plates, while only PBS 1× was placed in the other plate—so it could fit as a control plate—for 15 min at room temperature in order to evaluate the binding strength between the antibodies of the patients and the *Toxoplasma gondii* antigens.

The results of this test were expressed with the avidity index (AI) according to the formula AI = EI of the urea cavity/EI of the cavity without urea × 100. The interpretation for the results obtained for the percent avidity of the IgG1 and IgG antibody were as follows: 0–30%, low avidity—suggests that recent infection was acquired within the past three months; 31–59%, moderate avidity—suggests that the period of infection cannot be defined and is characterized as an indeterminate period; and 60–100%, high avidity—suggests that the infection was acquired more than three months ago [27].

### 2.5. Blood Collection and Isolation and Culture of Peripheral Blood Mononuclear Cells (PBMCs)

Venous blood (20 mL) was collected by venipuncture into heparinized tubes. PBMCs were isolated by Ficoll-Hypaque density gradient centrifugation (GE Healthcare, Uppsala, Sweden) at 430× *g* and 25 °C for 30 min. The cells were washed three times and resuspended in RPMI 1640 medium (GE Healthcare) containing 50 mM 4-(2-hydroxyethyl)-1-piperazineethanesulfonic acid (HEPES) buffer (Gibco), 10% inactivated fetal calf serum (Gibco), 2 mM L-glutamine (Gibco), 50 mM b-mercaptoethanol (Gibco), 24 mM sodium bicarbonate, and 40 ug/mL gentamicin (Neoquímica, Anápolis, GO, Brazil) to reach a final concentration of 2 × 106 cells/mL. PBMCs were cultured in 24-well microplates (Falcon, San Jose, CA, USA) in the presence of 5 ug/mL *T. gondii* antigens (strain RH), 1 µg/mL of αCD3 (BD Pharmigen^TM^, BD Biosciences, San Jose, California, USA), and 0.5 µg/mL of αCD28 (BD PharmigenTM, USA) or maintained in a culture medium at 37 °C in a humidified atmosphere with 5% of CO_2_. All procedures were performed under sterile conditions using a laminar flow hood. Supernatants were collected and stored at −80 °C for the quantification of cytokines.

### 2.6. Flow Cytometry

After the 48 h supernatant was collected, the PBMCs contained in each culture condition, αCD3αCD28, STAg, and absence of stimulation, from each patient were removed from the culture plate and divided into two microplate wells with U-bottom (SARSTEDT, Germany) diluted in PBS containing 10% AB serum (PBS-AB10%) to a final volume of 100 µL per well—the first containing unstained or blank cells (B) and the second containing cells to be stained. The PBMCs of each patient cultured for 96 h were previously incubated for five hours with Golgistop^TM^ solution (BD Biosciences, San Jose, CA, USA). Subsequently, the supernatant was collected, and the cells were removed from the culture plate and divided into five microplate wells with U-bottom, 100 µL per well diluted in PBS-AB10% and one well containing the blank, and the others containing the cells to be stained, for each of the three culture conditions. The extracellular probes made on cytometry after 48 h and the extra- and intra-cellular probes made after 96 h are presented in Table 1 below.

The protocol for cytometry analysis was the same as described in [28]. Briefly, the cells were resuspended in 150 µL of 4% paraformaldehyde fixative. Subsequently, the cells were passed into tubes to enable the reading on the BD FACSCantoTM II flow cytometer (BD Biosciences, USA) and visualization of the acquisition of 50,000 events per tube using the CellQuest software (BD Biosciences, USA). Antibodies that were appropriate for the control isotypes were used. The acquisition analysis (Appendix A) was performed using the FlowJo program, version 10.6.1 (BD Biosciences, USA).

### 2.7. Determination of Cytokines in the Culture Supernatant by Cytometric Bead Array (CBA)

The cytokines, IL-2, IFN-γ, IL-6, TNF-α, IL-4, IL-10, and IL-17, present in culture supernatants were simultaneously quantified by the CBA technique (BD ™ CBA Human Th1/Th2/Th17 Cytokine Kit, San Jose, CA, USA) according to the manufacturer’s protocol. Then, the beads bound with the cytokines were resuspended in 200 µL of the wash buffer and transferred to cytometry tubes for acquisition that was made on the same day on the FACSCalibur BD flow cytometer (BD Biosciences, USA). The acquisition analysis was performed using the FCAP Array^TM^ version 2.0 program (BD Biosciences, USA), and the concentration of cytokines was estimated by linear regression analysis with the fluorescence obtained on the standard curve of each cytokine and expressed in pg/mL.

For this analysis, 10 individuals from the CG, 10 patients from the SD group, and 9 from the IB group were selected, and the levels of cytokines of the three stimulus conditions, namely αCD3αCD28, STAg, and absence of stimulus, were measured.

### 2.8. Statistical Analysis

A statistical analysis of all data was performed using the GraphPad Prism software version 7.04 (GraphPad Software Inc., San Diego, CA, USA). The normal distribution of the quantitative variables was verified with the Shapiro–Wilk normality test [29,30]. Continuous variables that presented a normal distribution were expressed as mean ± standard deviation, and those with non-normal distribution were expressed in medians and interquartiles. The Kruskal–Wallis non-parametric test with Dunn’s post-test was used for data that did not follow normal distribution, and one-way ANOVA with Tukey’s post-test was employed when the distribution was normal. For the analysis of paired samples, the Wilcoxon test was used when the distribution was not normal and the paired *t*-test was employed for samples with normal distribution. Correlation analyses were performed using the Spearman correlation test. Values of *p* < 0.05 were considered statistically significant.

## 3. Results

### 3.1. General Characteristics of the Studied Individuals

The mean age among the tested individuals was 50.14 (±8.25) for the SD group, 48.38 (±11.39) for the IB group, and 38 (±10.00) for the CG. The other characteristics are described in Appendix A. The patients’ treatment schedules and medication use time are shown in Appendix A.

### 3.2. Detection of Total IgG Antibodies and Subclass IgG1 Anti-Toxoplasma gondii and Avidity of Total IgG and IgG1

Patients who had previous contact with *Toxoplasma gondii* were selected from the serology assessment for total IgG anti-*T. gondii*, as presented in Figure 2A. When the levels of total IgG were compared among the groups, there was no significant difference; however, a tendency to decrease in group IB compared to the CG was observed (*p* = 0.05) (Figure 2A). There were no significant variations in the evaluation of avidity, and all subjects presented antibodies with high avidity of total IgG, except for one patient in the IB group with moderate avidity (Figure 2B).

The presence of the IgG1 subclass, which is described in the literature as the most frequent in the anti-T. gondii response [31], was also assessed. The levels of serum IgG1 (Figure 2C) did not reveal statistically significant differences in the comparison among the groups; however, the IgG1 avidity test showed a reduction in avidity with significant difference in the IB group compared to the CG (*p* = 0.04) (Figure 2D). There was also a significant positive correlation between the total IgG and IgG1 levels in all groups, with *p* = 0.0003 and r = 0.86 in the CG, *p* < 0.0001 and r = 0.915 in the SD group, and *p* < 0.0001 and r = 0.97 in the IB group (Figure 2E).

### 3.3. CD4^+^ Expressing the Exhaustion Marker PD-1 and Activation Markers CD69^+^ and CD28^+^

PBMCs were cultured for 48 h and 96 h in the presence of medium alone or with STAg or αCD3αCD28 stimulation, respectively. Then, CD4^+^ and CD8^+^ cells were analyzed for the presence of activation or exhaustion markers. These results were analyzed for cell gain or loss in the stimuli used compared to cells that were not stimulated within the same group. A further analysis was conducted with the same stimulus between groups.

The expression of activation marker CD69^+^ in CD4^+^ lymphocytes demonstrated a significant increase in αCD3αCD28 after stimulation compared to the medium alone in the SD and IB groups (*p* = 0.0017 and *p* = 0.007, respectively). On the other hand, in the CG, there was only a tendency to increase in that same stimulus (*p* = 0.06). There was no difference after the STAg stimulation (Figure 3A). In CD8^+^ lymphocytes, the expression of CD69^+^, after αCD3αCD28 stimulation, was significantly increased in all groups compared to medium alone (*p* = 0.0007 in the CG, *p* = 0.0009 in the SD group, and *p* = 0.0003 in the IB group). STAg stimulation also promoted a significant increase in CD69 expression in all groups (*p* = 0.03 in the CG, *p* = 0.024 in the SD group, and *p* = 0.004 in the IB group). There was a significant low percentage of activated CD8^+^ cells in the absence of stimulus only in the IB group compared to the CG (*p* = 0.04) (Figure 3B).

CD4^+^ and CD8^+^ T cells were also analyzed for the expression of CD28^+^, a co-stimulatory molecule that interacts with B7 present in APC [32]. On CD4^+^ cells, the expression of CD28 was down-modulated in cultures treated with αCD3αCD28 when compared to the absence of stimuli in all groups (*p* = 0.0002 in CG, SD, and IB). The stimulation with STAg did not show significant changes (Figure 4A). Regarding CD8^+^CD28^+^ T lymphocytes, αCD3αCD28 presented a lower percentage than the medium in the CG (*p* = 0.014), SD (*p* = 0.0209), and IB (*p* = 0.015), and there were no significant changes in any groups in STAg. The cellular percentage in the stimulus with αCD3αCD28 in the IB group was reduced compared to the same stimulus in the CG (*p* = 0.03), and also in the stimulus with STAg in comparison between IB × CG (*p* = 0.03) (Figure 4B).

Cell exhaustion was also assessed by the expression of Programmed cell death protein 1 (PD-1). In CD4^+^, the increase in cells expressing this protein after stimulation with αCD3αCD28 was evident when compared to cells without any stimulus in the three groups (*p* = 0.0005 in the CG, *p* = 0.0012 in the SD group, and *p* < 0.0001 in the IB group). This increase in PD-1 expression was also observed in cells stimulated with STAg when compared to the absence of stimulus in the three groups (*p* = 0.0005 in the CG, *p* = 0.04 in the SD group, and *p* = 0.007 in the IB group). However, when the expression was compared among the groups, there was a statistical difference only between the non-stimulated cells of SD × CG (*p* = 0.008) and between the IB group and the CG (*p* = 0.0121) (Figure 4C). In CD8^+^ T lymphocytes, there was the same increase in expression of PD-1 in the cells stimulated with αCD3αCD28 when compared to the absence of stimulus in the three groups (*p* = 0.0002 in the CG, *p* = 0.0001 in the SD group, and *p* = 0.0003 in the IB group), as well as in the cells stimulated with STAg compared to the medium, although this increase occurred only in the CG (*p* = 0.003) and in the SD group (*p* = 0.0012). There were no statistically significant differences in CD8^+^ cells expressing PD-1 in the analysis between groups (Figure 4D).

Cells expressing both the CD28^+^ and PD-1 molecules were also analyzed. An increase in double expression was observed in CD4^+^ T lymphocytes stimulated with αCD3αCD28 when compared to unstimulated cells in all groups, with *p* = 0.03 for the CG, *p* = 0.04 for the SD group, and *p* = 0.0110 for the IB group. This same increase was also observed in the analysis of STAg × medium in the three groups: CG (*p* = 0.0005), SD (*p* = 0.04), and IB (*p* = 0.006). In the analysis among the groups, only the non-stimulated cells had a statistically significant difference between the SD group and the CG, demonstrating an increase in the percentage of cells with double expression in the SD group (*p* = 0.03) (Figure 4E).

The evaluation of CD8^+^ double positives demonstrated an increase in the percentage of these cells in the αCD3αCD28 stimulus compared to the medium in the CG (*p* = 0.0002) and SD (*p* = 0.0001) and IB (*p* = 0.0021) groups and in the STAg when compared to the medium in the CG (*p* = 0.003) and the SD group (*p* = 0.0105). A decrease was observed in the percentage of cells stimulated with STAg that presented this double staining in the IB group when compared to the CG (*p* = 0.04) (Figure 4F).

### 3.4. Intracellular Expression of IFN-γ, IL-17, and IL-10 Cytokines by CD4^+^ Cells

CD4^+^ T cells were also evaluated for their expression of IFN-γ, IL-17, and IL-10 and double expression of these cytokines. IFN-γ producing cells had an increase in their percentage after stimulation with αCD3αCD28 compared to those not stimulated in the CG (*p* = 0.0002) and SD (*p* = 0.03) and IB (*p* = 0.0013) groups. This same increase in comparison to the medium was observed after stimulation with STAg in the CG (*p* = 0.0002) and in the IB group (*p* = 0.0013) but not in the SD group. The analysis among the groups did not reveal significant differences (Figure 5A).

The percentage of CD4^+^IL-17^+^ in the cells stimulated with αCD3αCD28 was higher than those that were not stimulated in the CG (*p* = 0.006) and in the IB group (*p* = 0.0016). The same occurred in the cells after stimulation with STAg (*p* = 0.0215 in the CG and *p* = 0.03 in the IB group). However, there were no differences in the stimuli in the analysis among the groups surveyed (Figure 5B).

The intracellular expression of IL-10 was increased in CD4^+^ cells after stimulation with αCD3αCD28 compared to the medium in groups CG (*p* = 0.013) and IB (*p* = 0.007) and after stimulation with STAg compared to the medium in the CG (*p* = 0.03) and SD (*p* = 0.024) and IB (*p* = 0.025) groups (Figure 5C); there was no difference among the groups.

With regard to the double intracellular expression of cytokines, the evaluation of CD4^+^IFN-γ^+^IL-17^+^ demonstrated an increase in the percentage of cells stimulated with αCD3αCD28 and STAg compared to unstimulated lymphocytes in the CG (*p* = 0.006 and *p* = 0.03, respectively) and IB group (*p* = 0.006 and *p* = 0.029, respectively). However, there were no differences in the assessment among the groups (Figure 5D). In turn, double cells producing IFN-γ^+^ and IL-10 presented an increase in their percentage only in the CG when αCD3αCD28 and STAg were compared with unstimulated cells (*p* = 0.008 and *p* = 0.013, respectively), while in the IB group, there was only a tendency to increase the percentage of cells stimulated with STAg in comparison to the medium (*p* = 0.073). However, the analysis among the groups did not show any significant difference (Figure 5E). Finally, the CD4^+^IL-17^+^IL-10^+^ lymphocytes demonstrated a significant increase in the CG only when the cells stimulated with αCD3αCD28 were compared to non-stimulated cells (*p* = 0.0105). They also showed a tendency to increase in STAg × medium in the CG (*p* = 0.05) and αCD3αCD28 × medium in the IB group (*p* = 0.07). Similarly, there were no significant changes when the differences among the groups in each stimulus were analyzed (Figure 5F).

### 3.5. Assessment of Regulatory CD4^+^ T Populations

The profile of regulatory T lymphocytes was also analyzed. CD4^+^ T cells were evaluated for the presence of the alpha subunit of the receptor of the interleukin-2 or CD25, the Foxp3 transcription factor, and the latency-associated peptide (LAP). During the cytometry analysis, cells were separated into CD25High, CD25Low, and CD25 negative, being considered as classic Tregs, stained as CD4^+^CD25HighFoxP3^+^.

An increase in CD4^+^CD25High T lymphocytes was observed in the stimuli: αCD3αCD28 compared to cells without any stimulus in the three groups (*p* = 0.0002 in the CG, *p* = 0.0001 in SD group, and *p* < 0.0001 in the IB group), as well as in STAg when compared to the medium in groups CG, SD, and IB (*p* = 0.0002, *p* = 0.04, and *p* = 0.0004, respectively). When the difference among the groups was analyzed, the percentage of these cells was reduced after being stimulated with STAg in the SD group compared to the CG (*p* = 0.017); however, there were no differences in the IB (Figure 6A).

CD4^+^CD25HighFoxp3^+^ cells had an increase in their percentage in the stimulus with αCD3αCD28 compared to the medium only in the IB group (*p* = 0.03) and did not present significant changes in the analysis among the groups (Figure 6B). In turn, CD4^+^CD25HighLAP^+^ cells had an increase in their percentage when stimulated with STAg compared to those without any stimulus in all groups (*p* = 0.0105 in the CG, *p* = 0.04 in the SD group, and *p* = 0.006 in the IB group) and they had no gain or loss in any stimulus when compared among the groups (Figure 6C). However, regulatory T cells expressing Foxp3^+^ and LAP^+^ did not present significant differences between the stimuli of the groups but only a tendency to increase in STAg compared to the medium in group IB (*p* = 0.07); there was no difference in the analysis among the groups (Figure 6D).

### 3.6. Production of Cytokines by PBMCs in Patients with Autoimmune Rheumatic Diseases

The supernatant was analyzed for the presence of the cytokines IL-2, IFN-γ, IL-6, TNF-α, IL-4, IL-10, and IL-17 after 96 h of cell culture. In the analysis of IL-2 production, an increase in this cytokine was observed in STAg when compared to the absence of stimulation in the three groups, with *p* = 0.003 in the CG and in the SD group and *p* = 0.019 in the IB group (Figure 7A). The next cytokine that was analyzed was IFN-γ, which demonstrated a significant increase in its production in αCD3αCD28 and STAg when compared to the medium in all groups (*p* = 0.002 in the CG and SD group and *p* = 0.003 in the IB group for both stimuli), as shown in Figure 7B.

With regard to IFN-γ, IL-6 demonstrated an increase in its production in both stimuli when compared to its production in unstimulated cells in all groups. The significant differences for αCD3αCD28 were *p* < 0.0001 for the CG and SD group and *p* = 0.0001 for the IB group, while for STAg, the significant differences were *p* = 0.0004, *p* = 0.004, and *p* = 0.026 for the CG, SD group, and IB group, respectively (Figure 7C). TNF-α, in turn, increased its production after stimulation of cells with αCD3αCD28 compared to the medium in the SD and IB groups (*p* = 0.007 for both) and showed a tendency to increase its production in the CG (*p* = 0.07). The STAg stimulation also resulted in an increase in this cytokine in the three groups, with *p* = 0.002 in the CG, *p* = 00.78 in the SD group, and *p* = 0.003 in the IB group (Figure 7D).

The analysis of IL-4 had no statistically significant differences between stimuli when compared with cells that were not stimulated in any of the groups (Figure 7E). On the other hand, IL-10 presented a significant increase in its production after stimulation with αCD3αCD28 when compared to the absence of stimuli in the three groups: CG and SD group (*p* = 0.002) and IB group (*p* = 0.003). However, only the IB group demonstrated an increase in the production of this cytokine in STAg compared to the medium (*p* = 0.007) (Figure 7F).

Finally, IL-17 was increased only in αCD3αCD28 when compared to the medium, significantly statistically in the three groups (*p* = 0.007 in the CG and SD group and *p* = 0.015 in the IB group), as demonstrated in Figure 7G.

The analysis of cytokine production in the same stimulus among the study groups was also carried out, but no increase or decrease in any of the cytokines surveyed was observed (Figure 7A–G).

## 4. Discussion

An appropriate response against *Toxoplasma gondii* depends on the action of cytokines, such as IL-12, IFN-γ, and TNF-α, which are predominantly of the Th1 profile, in addition to other pro-inflammatory cytokines [33]. In turn, the control of autoimmune diseases requires the reduction in the excessive action of pro-inflammatory cytokines in individuals affected by these pathologies. In this context, TNF blockers are considered a great advance in the fight against autoimmune diseases, contributing to their control and providing a good quality of life for patients in most cases [34]. However, there are some cases in which the medication may not work as expected (therapeutic failure) [35] or may work but cause an immunosuppression that predisposes the individual to opportunistic diseases [18,19]. Previously in our laboratory, it was demonstrated that the use of infliximab, a TNF-α blocker, in an in vitro granuloma model using PBMCs from patients with active or treated tuberculosis or positive PPD, was able to promote a negative modulation of Treg, Th1, and Th17 profiles, which was evaluated through the observation of decreased production of cytokines IFN-γ, IL-12p40, IL-10, and IL-17 [15]. Thus, given the importance of a balance in the immune responses of individuals to avoid tissue damage or involvement by opportunistic diseases, this study sought to verify whether, as has already been shown in tuberculosis, the use of these immunosuppressive drugs in carrier patients of autoimmune diseases impairs their ability to respond against *Toxoplasma gondii*.

There are reports of individuals who can develop neurological or ocular symptoms [18,19,36]; however, this is the first study, as far as we know, that tried to establish and discover which immunological changes are related to a possible reactivation. Moreover, recently, our study group showed that pregnant women with gestational diabetes had lower anti-*T. gondii* than those without diabetes. They also had a higher number of T lymphocytes expressing activation and exhaustion markers (CD28^+^ and PD-1), a lower number of CD4^+^ T cells producing IFN-γ, IL-10, and IL-17, and lower secretion of IL-17, IL-4, TNF, and IL-2 after their PBMCs are challenged in vitro with *T. gondii* antigens, which indicates the importance of monitoring patients with immunosuppression, even if it is transient [37]. For this reason, we try to evaluate some parameters of cell expression and cytokine production in vitro after nonspecific stimulation and a stimulation with total soluble *Toxoplasma gondii* antigens.

In our study, we found that CD4^+^ cells expressing the PD-1 exhaustion marker were increased in patients using synthetic and biological drugs compared to healthy individuals; however, this exhaustion was only observed in the absence of stimuli. It has been demonstrated that the increase in PD-1 occurs in response to the exhaustion in T lymphocytes [38,39]. For this reason, PD-1 is also involved in autoimmunity and peripheral tolerance mechanisms, and there are studies associating it with the development of therapeutic approaches to alleviate the effects of some autoimmune diseases [39]. We found that this change confirms its relationship of increased expression in autoimmune diseases, as described in the literature [39], since the cells of these patients may present signs of exhaustion due to being constantly activated. The increase in its expression may also be due to the attempt to decrease the activation of T cells in order to make them less reactive, thus reducing the inflammatory effects observed in autoimmune diseases [40].

There is also evidence that chronic toxoplasmosis infection can promote cell exhaustion and increase PD-1 expression mainly in the presence of bradyzoites, as demonstrated in mice [41,42,43]. In fact, when we analyzed the cells stimulated with STAg compared to the non-stimulated ones, we were able to observe an increase in CD4^+^ exhaustion in the three groups and in CD8^+^ in CG and SD, although this change also occurred after exposure to the mitogen in the three groups for both cell populations.

One study revealed that lymphocyte choriomeningitis virus-specific CD8^+^ cells presented increased expression of this receptor accompanied by impaired proliferation and decreased production of cytokines in effector cells. One of its ligands, PD-L1, was also increased in splenocytes from infected mice. Moreover, blocking PD-L1 with antibodies led to an increase in responses in these cells again, including an increase in the production of cytokines TNF-α and IFN-γ and a decrease in the viral load in organs of these animals even in the absence of CD4^+^ lymphocytes. Anti-PD-L1 also allowed an increase in the proliferation of this population since it does not allow a communication between PD-1 and its ligand, as it still remains expressed on the cell surface. Blocking PD-1 itself is able to improve CD8^+^ exhaustion again, although to a lesser extent [38]. In toxoplasmosis, PD-1 expression in CD8^+^ T cells has been associated with increased apoptosis and decreased proliferation [42]. Furthermore, mice infected with *T. gondii* and in the chronic phase that presented antibodies against MAG1—which is an antigen present in the cyst containing bradyzoites—expressed higher levels of PD-1 and its two ligands, namely PD-L1 and PD-L2. It was seen that the greater the number of cysts, the higher the antibody levels and the greater the expression of these molecules. However, once treated with anti-PD-L1, the mice showed a decrease in the number of brain cysts containing bradyzoites and decreased expression of the BAG1, a protein which is also present in bradyzoites. Another positive point was that this blockade promoted an increase in cytokines, such as IL-12p70 and IL-10, which were correlated with a decrease in cysts [43].

Here, we observe that PD-1 expression was equally modulated in the CG and SD and IB groups in CD4^+^ or CD8^+^ after αCD3αCD28 and STAg stimuli. This finding suggests that these cells may present the capacity of being responsive when stimulated, although they were obtained from patients who were suffering from an autoimmune disease and who were under treatment. Given the significance of the PD-1 PD-1L signaling pathway in homeostasis preservation and in the anti-infectious response, it is important to note that according to our results, treatment with TNF blockers does not represent a potential risk to modulate PD-1 expression.

Furthermore, CD4^+^ cells with simultaneous expression of CD28 and PD-1 were increased in the absence of SD stimuli, confirming once again that the presence of PD-1 did not completely inhibit the presence of CD28 and a possible activation via CD28-B7. This may be because although PD-1 can interfere with CD28-mediated PCK-ϴ activation and decrease T cell activation, unlike CTLA-4, it does not exclude the presence of this molecule. It is not positioned in a similar location and therefore does not compete with CD28 [40]. Thus, it is possible for a cell to express both molecules even though PD-1 indirectly impairs the proper function of CD28.

When it comes to Tregs, we know that the TNF-α cytokine is capable of compromising the proper function of this population in rheumatoid arthritis, and one of the mechanisms is to reduce the phosphorylation of the FoxP3 transcription factor so that the treatment with anti- TNF contributes to the reestablishment of the function of these cells, reversing this effect and further reducing the action of IFN-γ and IL-17 and promoting a balance between cell profiles [44].

Due to the use of TNF blockers, which, despite being one of the main cytokines causing inflammation in autoimmune diseases, are also one of those responsible for maintaining the anti-*T. gondii*, we thought that there could be a reduction in the percentage of cells expressing IFN-γ and an increase in regulatory T profile cells; however, this did not occur. Not only the cells expressing IFN-γ were able to present similar percentages in the study groups and in the CG but there was also a decrease in the CD4CD25High population in patients using synthetic drugs after stimulation with STAg when compared to cells from individuals of the CG. This suggests that it might be possible to develop an appropriate Th1 response in these patients in face of a possible infection by the parasite. However, when the transcription factors FoxP3 and LAP were analyzed in this population, there were no significant differences. Hence, it is not possible to affirm that all these cells in fact belonged to the regulatory profile.

Contrary to our results, Yang et al. (2020) found that the percentage of regulatory T cells was lower in patients with ankylosing spondylitis than that in the CG; however, after patients were treated with anbainuo, which is a recombinant TNF-α receptor (biosimilar etanercept), for 12 weeks, they found that the medication promoted an increase in the number of these cells [45]. This was also seen in a murine model of arthritis, in which TNF blockade was accompanied by an increase in this population [46]. The fact that we found a decrease rather than an increase in Tregs may suggest that the effects of the anti-TNF assessed did not stand out from an adequate response to the stimuli. During *T. gondii* infection, there is a large production of IFN-γ, expression of *T*-bet, and an increase in Th1 cells; therefore, there is a reduction in Foxp3^+^ Tregs, which should be balanced to avoid the inflammatory effects caused by defense against the parasite to increase the pathogenesis of the disease [47]. Another explanation for this reduction in Tregs would be that, for some reasons, these patients did not present the maximum effects of immunobiologicals since even in the absence of stimulus, the Tregs were reduced. However, it is noteworthy that this change in regulatory T cells may or may not occur depending on how the disease progresses in these individuals and even on the type of TNF blocker used for treatment. Patients using adalimumab, for example, may have increased Tregs, but if the disease activity persists despite immunobiologicals, the increase in those cells will not be detected [48]. In addition, there is variation in the immune response according to the type of TNF blocker used, since this increase in Tregs due to the use of adalimumab does not occur in the treatment with etanercept [48].

Finally, when we evaluated the expression of cytokines IFN-γ, IL-17, and IL-10 through flow cytometry, we noted that stimulation with STAg and αCD3αCD28 did not increase or decrease the capacity to express cytokines in any of the groups, which indicates that the use of immunomodulating medication, whether synthetic or biological, still allows the response to be similar to that observed in individuals without autoimmune disease, such as those who composed the CG. Accordingly, an analysis of peripheral blood CD4^+^ and CD8^+^ T cells from patients with rheumatoid arthritis, placed in culture and stimulated with PMA and ionomycin or with αCD3αCD28 in the presence of brefeldin A, verified the expression of Th1 and Th2 profile cytokines IL-2, IFN-γ, IL-4, IL-5, IL-13, and IL-10 in these cells and concluded that the percentages of expression of these cytokines were very similar in patients with rheumatoid arthritis compared to controls without any type of autoimmune disease [49]. Just as we evaluated possible differences between cytokines in patients using immunobiologicals (IB), synthetic drugs that are also immunosuppressive (SD), and the CG, this study also evaluated these possible differences between patients receiving immunosuppressants and those receiving only nonsteroidal anti-inflammatory drugs and also did not find significant differences in cytokine expression [49].

Regarding cytokine analyses in the culture supernatant, Sauzullo et al. (2018) examined the production of the cytokine IFN-γ in response to stimulation with the mitogen phytohemagglutinin (PHA) in patients with rheumatologic immune-mediated inflammatory diseases (IMID) for one to eight years of prolonged use of TNF-α blockers. The study discovered that over these years, there were fluctuations in the production of IFN-γ, with the lowest levels in the first dose and after four years of use, while the doses after one, two, and eight years were even larger than the initial one, and there were no differences between the production of this cytokine in the two TNF blockers evaluated, namely etanercept and adalimumab, or between the two diseases, rheumatoid and psoriatic arthritis. It is also interesting to note that after these individuals used these immunobiologicals for eight years, no statistical differences were found in the production of IFN-γ in them when compared to healthy donors [50]; this is similar to our study, which also did not find differences in the production of any of the analyzed cytokines, IL-2, IFN-γ, IL-6, TNF-α, IL-4, IL-10, and IL-17, in the supernatant in IB and SD groups when compared to the CG, neither in the stimulation with the mitogen nor in the stimulus with *T. gondii* antigen.

Another study [51] evaluated the effects of an anti-TNF after seven days of drug application and, contrary to our results, the researchers found that TNF production after stimulation with αCD3αCD28 was lower in patients without treatment, patients treated with methotrexate, and patients using anti-TNF compared to the CG. Additionally, the production of IL-10 and IL-6 was lower in patients treated with methotrexate or biological drugs. They also found that CD4^+^CD69^+^ cells were decreased in individuals with MTX and IB after stimulation with αCD3αCD28 [51], while our study did not indicate alterations in the CD4^+^ population but only in CD8^+^. However, as provided in our study, Furiati et al. (2019) did not observe differences in the production of IFN-γ and IL-17 when comparing the CGs, patients with untreated psoriasis, and patients with psoriasis being treated with synthetic or biological DMARDs [51].

These findings of cell expression of cytokines and cytokine production demonstrating similar levels between individuals using immunosuppressive drugs and the CG, although preliminary, are surprisingly encouraging, since we can suggest, at least initially, that the risk of toxoplasmosis reactivation is not as increased as that observed for the reactivation of tuberculosis, for example. In fact, the changes caused by the autoimmune disease alone are enough to increase the risk of infections in these patients by at least twice, including tuberculosis [9]. In addition, when the treatment is conducted with immunobiologicals, this risk increases even more, having already been exposed that, depending on the type of blocker used, it can reach about 30 times, as presented by Seong et al. (2007) regarding the use of infliximab [52]. However, there are few studies that relate the use of biological therapies to the risk of toxoplasmosis reactivation, and, as far as we know, there are no other studies presenting details of the immune responses of these individuals to stimuli with *T. gondii* antigens; therefore, we would not be able to establish what degree of immunosuppression would be necessary for a possible reactivation based on the results observed here.

However, it is known that in individuals with AIDS, who have an immunosuppression that is much more pronounced than the immunosuppression caused by DMARDs used by patients with autoimmune diseases, the risk of neurological disorders triggered by the reactivation of tuberculosis is much greater than the risk of attacks caused by neurotoxoplasmosis; therefore, while tuberculous meningitis is associated with CD4^+^ levels below 400/mm^3^, toxoplasmosis is associated with CD4^+^ levels below 200/mm^3^ [53]. However, even HIV-positive patients can use immunobiologicals to treat autoimmune diseases. Some studies demonstrate that immunosuppressants could be administered to these individuals as long as they are well-monitored. The number of viral copies, as well as the CD4^+^ cell count, can be useful for proper follow-up [54,55]. A review published in 2016 by Gallitano et al. described that of 27 cases of patients treated with TNF blockers, only 4 had complications due to infections [55]. Another study followed HIV-positive patients with rheumatic disease using TNF inhibitors, between 2003 and 2021, and observed that in general, these drugs did not cause serious infectious episodes, being considered relatively safe even in the long term [56]. This reaffirms the need for a balance between monitoring and treating individuals using this medication.

Therefore, it is possible to understand why, although there are reports of neurological and ocular manifestations in patients using anti-TNF and other immunobiologicals, there is still no monitoring or screening for toxoplasmosis during treatment, since at first, it seemed that other factors might be involved in this imbalance in immunity that caused reactivation in some of these patients.

Another fact to consider is that there are different responses to different types of TNF blockers. In cultures of cells from healthy individuals stimulated with M. tuberculosis antigens or PHA and incubated with TNF blockers, it was shown that both adalimumab and infliximab were able to inhibit the early activation of T cells, evaluated by the expression of CD69^+^, while etanercept was not able to do so. Furthermore, it was observed that TNF blockade with infliximab and adalimumab could suppress IFN-γ production, while again, etanercept did not differ from non-blocked cells [57]. It is also necessary to consider that the immune response, even in individuals without autoimmune diseases, may vary from one another. In this context, autoimmune disease is another confounding factor, since even when evaluating the use of only one type of immunobiological intervention, one must keep in mind that this medication is used for several autoimmune diseases with different pathophysiologies. There is also the issue of other associated comorbidities that can interfere with the assessment (although we have tried our best to reduce this bias), and finally, it must be considered that an evaluation with a broader cell staining panel and a larger n is necessary.

## 5. Conclusions

To our knowledge, this is the first study that analyzed cellular immune response in patients with rheumatic autoimmune diseases and serum positive to toxoplasmosis under the treatment of TNF blockers. The absence of significant changes in the analyzed immune response suggests that these patients have a low risk of reactivation of toxoplasmosis.

## Figures and Tables

**Figure 1 biomedicines-11-00930-f001:**
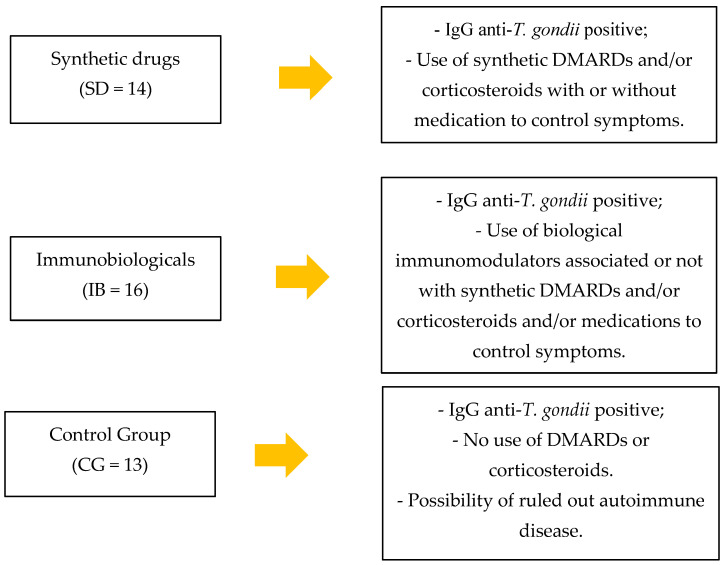
Scheme of division of groups.

**Figure 2 biomedicines-11-00930-f002:**
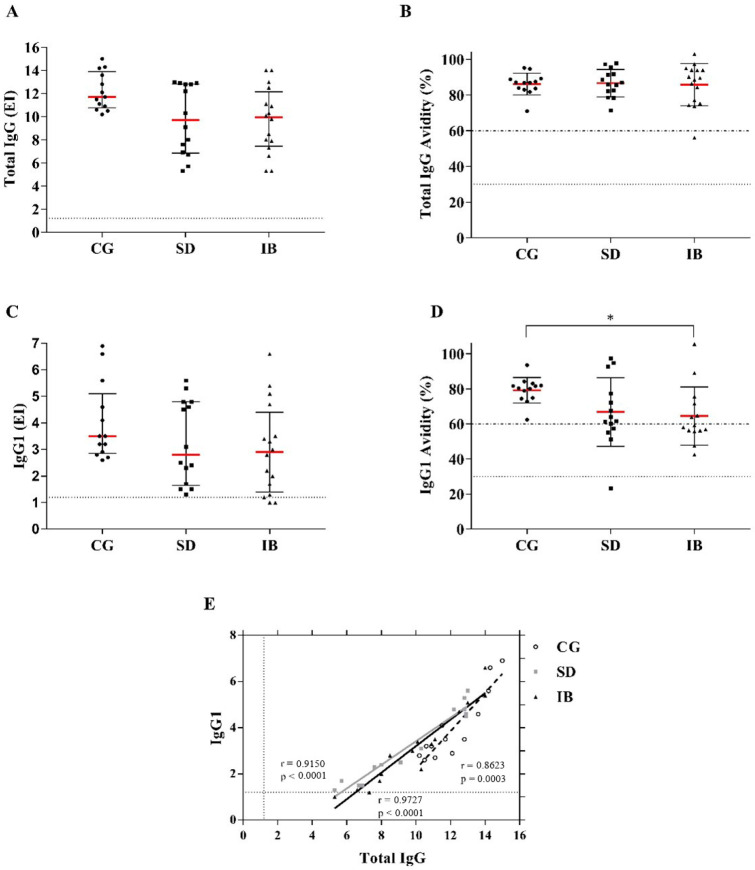
Evaluation of humoral immunity, assessed by the production of specific antibodies to Toxoplasma gondii. (**A**) Levels of total IgG anti-*T. gondii* antibodies, expressed in ELISA index (IE). (**B**) Avidity index of total IgG anti-*T. gondii* antibodies, expressed as a percentage (%). (**C**) Level of the IgG1 subclass anti-*T. gondii*, expressed in ELISA index (IE). (**D**) Avidity index of the IgG1 subclass anti-*T. gondii*, expressed as a percentage (%). (**E**) Correlation between the levels of total IgG antibodies and the IgG1 subclass, in groups CG, SD, and IB. Analysis among the groups by Kruskal–Wallis (**A**,**C**) and by ordinary one-way ANOVA (**B**,**D**), and correlation analysis by Spearman test, where r expresses the correlation coefficient. The horizontal bars represent the medians with interquartile ranges or means with standard deviation. Statistically significant data, * *p* < 0.05.

**Figure 3 biomedicines-11-00930-f003:**
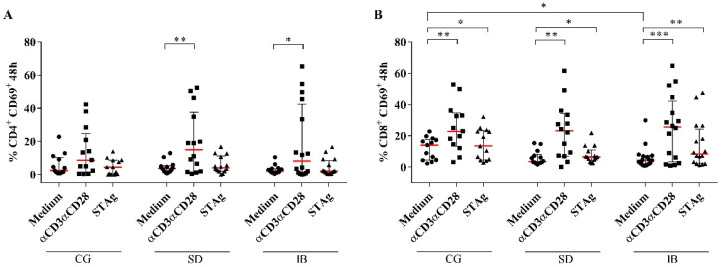
Activation of CD4^+^ and CD8^+^ T lymphocytes, assessed by the expression of the CD69 cell surface receptor. (**A**) Percentage of total CD4^+^ and (**B**) CD8^+^CD69^+^ after 48 h of stimulus with anti-CD3 and anti-CD28, STAg, and absence of stimulus in the CG and SD and IB groups by flow cytometry. Analysis between stimuli by Wilcoxon and among groups by Kruskal–Wallis. The horizontal bars represent the medians with interquartile ranges. Statistically significant data, * *p* < 0.05, ** *p* < 0.005, and *** *p* < 0.0005.

**Figure 4 biomedicines-11-00930-f004:**
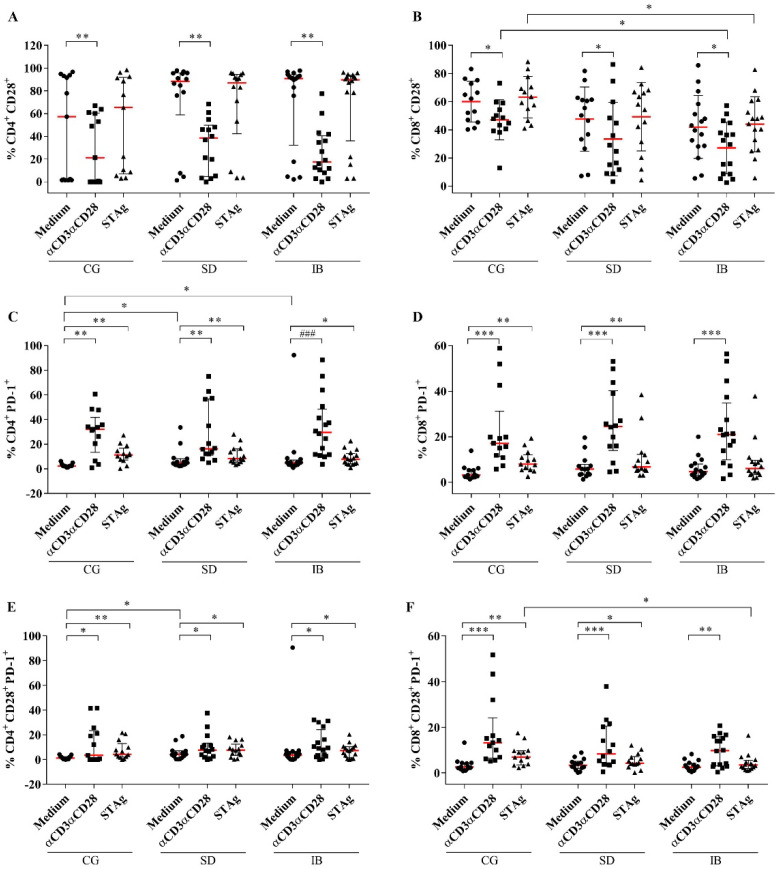
Activation and exhaustion of CD4^+^ and CD8^+^ T lymphocytes, assessed by the expression of CD28 and PD-1 cell surface receptors. (**A**) Percentage of CD4^+^CD28^+^, (**B**) CD8^+^CD28^+^, (**C**) CD4^+^PD-1^+^, (**D**) CD8^+^PD-1^+^, (**E**) CD4^+^CD28^+^ PD-1^+^, and (**F**) CD8^+^CD28^+^PD-1^+^ after 96 h of stimulus with anti-CD3 and anti-CD28, STAg, and absence of stimulus in the CG and SD and IB group, by flow cytometry. Analysis between stimuli by Wilcoxon (**A**,**C**–**F**) and by paired *T*-test (**B**) and among groups by Kruskal–Wallis (**A**,**C**–**F**) and ordinary one-way ANOVA (**B**). The horizontal bars represent the medians with interquartile ranges or means with standard deviation. Statistically significant data, * *p* < 0.05, ** *p* < 0.005, *** *p* < 0.0005, and ### *p* < 0.0001.

**Figure 5 biomedicines-11-00930-f005:**
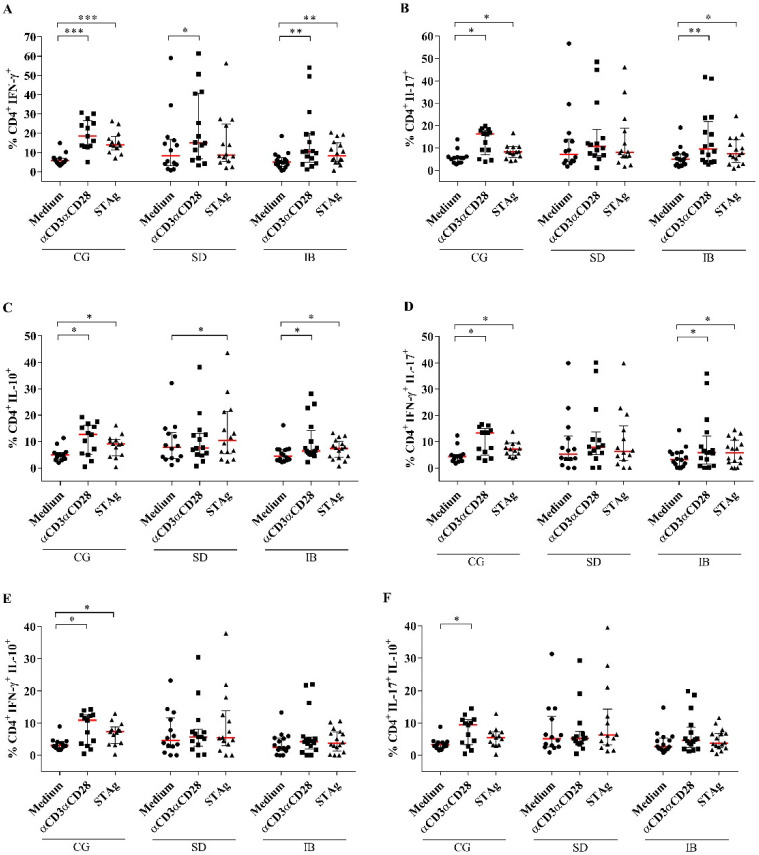
CD4^+^ T lymphocyte profiles, assessed by the expression of IFN-γ, IL-17, and IL-10 cytokines. (**A**) Percentage of CD4^+^IFN-γ^+^, (**B**) CD4^+^IL-17^+^, (**C**) CD4^+^IL-10^+^, (**D**) CD4^+^IFN-γ^+^IL-17^+^, (**E**) CD4^+^IFN-γ^+^IL-10^+^, and (**F**) CD4^+^IL-17^+^IL-10^+^ after 96 h of stimulus with anti-CD3 and anti-CD28, STAg, and absence of stimulus in the CG and SD and IB groups, by flow cytometry. Analysis between stimuli by Wilcoxon and among groups by Kruskal–Wallis. The horizontal bars represent the medians with interquartile ranges. Statistically significant data, * *p* < 0.05, ** *p* < 0.005, and *** *p* < 0.0005.

**Figure 6 biomedicines-11-00930-f006:**
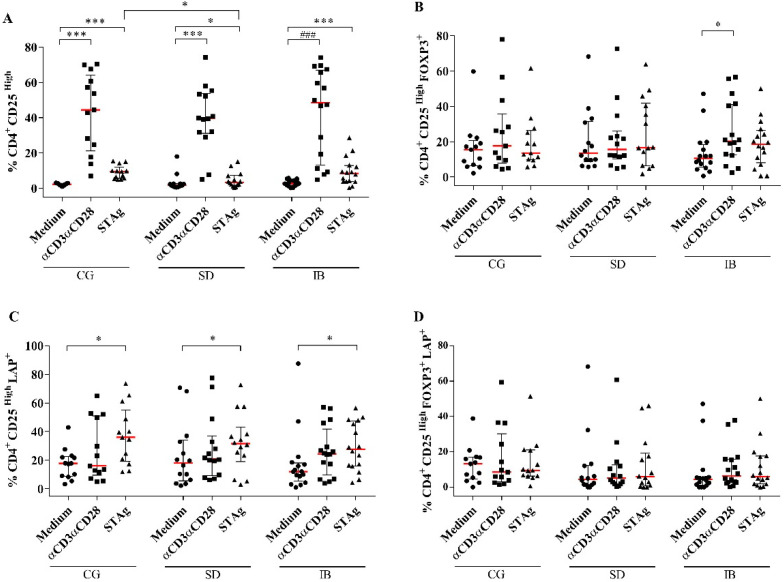
Regulatory CD4^+^ T lymphocytes profile, evaluated by the expression of the alpha chain of the IL-2 receptor (CD25), LAP, and the transcription factor Foxp3. (**A**) Percentage of the total CD4^+^CD25High, (**B**) CD4^+^CD25High Foxp3^+^, (**C**) CD4^+^CD25High LAP^+^, and (**D**) CD4^+^CD25High Foxp3^+^LAP^+^ after 96 h of stimulus with anti-CD3 and anti-CD28, STAg, and absence of stimulus in the CG and SD and IB groups, by flow cytometry. Analysis between stimuli by Wilcoxon and among groups by Kruskal–Wallis. The horizontal bars represent the medians with interquartile ranges. Statistically significant data, * *p* < 0.05, *** *p* < 0.0005, and ### *p* < 0.0001.

**Figure 7 biomedicines-11-00930-f007:**
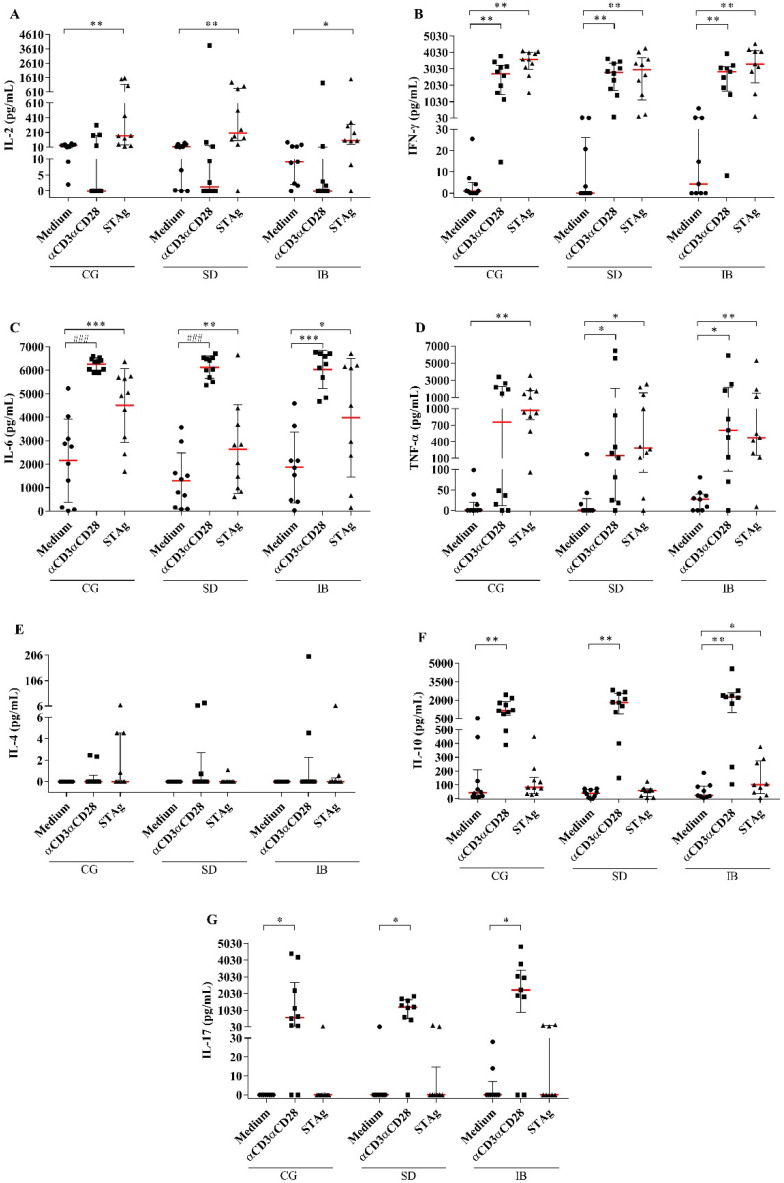
Cytokine production in cell culture supernatant. Cytokine levels of (**A**) IL-2, (**B**) IFN-γ, (**C**) IL-6, (**D**) TNF-α, (**E**) IL-4, (**F**) IL-10, and (**G**) IL-17, expressed in pg/mL, quantified after 96 h of stimulation with anti-CD3 and anti-CD28, STAg, and absence of stimulus in groups CG, SD, and IB, by CBA. Analysis between stimuli by Wilcoxon (except in C, paired *T*-test) and among the groups by Kruskal–Wallis (except in C, ordinary one-way ANOVA). The horizontal bars represent the medians with interquartile ranges or means with standard deviation. Statistically significant data, * *p* < 0.05, ** *p* < 0.005, *** *p* < 0.0005, and ### *p* < 0.0001.

**Table 1 biomedicines-11-00930-t001:** Stains used in the cytometry protocol.

Wells/Tubes	Culture Time	Extracellular Labeling *	Intracellular Labeling *
B	48 h	-----	-----
1	48 h	CD8 BB515	-----
		CD69 PE	
		CD4 PE-Cy7	
B	96 h	-----	-----
2	96 h	CD8 BB515	-----
		PD1 PE (CD279)	
		CD4 PE-Cy7	
		CD28 APC	
3	96 h	CD25 FITC (IL-2RA)	-----
		-----	FOXP3 PE
		CD4 PE-Cy7	-----
		LAP Alexa 647	-----
4	96 h	-----	IL-17 Alexa 488
		-----	IL-10 PE
		CD4 PE-Cy7	-----
		-----	IFN Alexa 647

* The antibodies used are BD PharmigenTM, USA. BB515, bright blue 515; PE, phycoerythrin; PE-Cy7, phycoerythrin cyanine 7; APC, allophycocyanin; FITC, fluorescein isothiocyanate; Alexa 647, Alexa Fluor 647; and Alexa 488, Alexa fluor 488.

## Data Availability

Not applicable.

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
