# Peer review of "Evaluation of Specific Cellular and Humoral Immune Response to Toxoplasma gondii in Patients with Autoimmune Rheumatic Diseases Immunomodulated Due to the Use of TNF Blockers"

_biomedicines, 2023, doi:10.3390/biomedicines11030930_

Round 1

Reviewer 1 Report

 In this observation, authors tried to verify whether patients with autoimmune rheumatic diseases with previous contact with Toxoplasma gondii (IgG+) have any indication of an increased risk of toxoplasmosis reactivation during the treatment with TNF antagonists and/or synthetic drugs. They concluded that immunobiologicals alone did not promote an immunological imbalance to the point of impairing the anti-Toxoplasma gondii immune response and predisposing to toxoplasmosis reactivation. From clinical point of view, this is of great importance and interesting observation in the field, which adds valuable clinical evidence at the cellular level and well definitely benefit for the clinical management of autoimmune rheumatic diseases with previous Toxoplasma gondii infection. The methodology used in this study is sound; the statistical analysis is appropriate; the conclusion is supported by their results. However, some issues need to be addressed in their revision.

 Detail comments:

1.     The description to their data in the result section is kind of repeatedly, so, the English writing needs to be reinforced and strengthened carefully;

2.     The figure 1 needs to be rearranged because the distance between the subfigure should be more closer like their other figure arrangement;

3.     Regarding how to decide the effective number after decimal point “.”, for example, the line-305, p = 0.0554, in this case, if the first number appeared can be divided by three (> or = 3), then the effective number should be described as p = 0.05, the 2nd “5” would be not necessary, in other word, it is ineffective; if the first number cannot be divided by three (< 3), so, the effective number would be two digit, for example, p = 0.017, “1” cannot be divided by 3, then the next digit “7” has to be included;

4.     There are several places need to be modified carefully as following:

(1)   Line-551: “increased” should be increase;

(2)   Line-556: in the sentence two “show” need to be avoided, please modify the it;

(3)   Line-564 & 565: CD8+ should be CD8+;

(4)   Line 579: “to consider” is better to change into to be considered;

(5)   Line-581: it is more appropriate to replace “tried” using “tended”;

(6)   Line-579: this sentence has to be modified accordingly simply because the these is not consistent;

(7)   Line-582: capable to do something should be capable of doing something;

(8)   Line-600: in this sentence, “to do” and “doing” should be consistent one it shares one subject;

(9)   Line-616: it would be more appropriate to use then that in the CG to instead of than in the CG;

(10)   Line-621: “suggest that ……” has to be use the present tense;

(11)   Line-657: please remove one “that”;

(12)   Line-706: “have already had contact” should be changed into have already been contacted ……;

Author Response

Response to Reviewer 1 Comments

Point 1:  The description to their data in the result section is kind of repeatedly, so, the English writing needs to be reinforced and strengthened carefully;

Response 1:

à We summarize part. of the results. However, as the study covers two types of analyses, between groups and between stimuli in the same group, we think it is important to emphasize the type of result mentioned.

Point 2: The figure 1 needs to be rearranged because the distance between the subfigure should be more closer like their other figure arrangement

Response 2: The adaptation was made in the figure, but a new figure was added in item 2.1 and was named as figure 1. Consequently, the others were renumbered according to the order in which they appear.

Point 3:  Regarding how to decide the effective number after decimal point “.”, for example, the line-305, p = 0.0554, in this case, if the first number appeared can be divided by three (> or = 3), then the effective number should be described as p = 0.05, the 2nd “5” would be not necessary, in other word, it is ineffective; if the first number cannot be divided by three (< 3), so, the effective number would be two digit, for example, p = 0.017, “1” cannot be divided by 3, then the next digit “7” has to be included.

 Response 3: The adaptation was made in the numbers after decimal point.

Point 4: There are several places need to be modified carefully as following:

(1)   Line-551: “increased” should be increase à OK

(2)   Line-556: in the sentence two “show” need to be avoided, please modify the it à OK

(3)   Line-564 & 565: CD8+ should be CD8+ à OK

(4)   Line 579: “to consider” is better to change into to be considered à OK

(5)   Line-581: it is more appropriate to replace “tried” using “tended” à OK

(6)   Line-579: this sentence has to be modified accordingly simply because the these is not consistent; à OK

 (7)   Line-582: capable to do something should be capable of doing something à OK

(8)   Line-600: in this sentence, “to do” and “doing” should be consistent one it shares one subject;

à We did not find the reported inadequacy.

(9)   Line-616: it would be more appropriate to use then that in the CG to instead of than in the CG à OK

(10)   Line-621: “suggest that ……” has to be use the present tense à OK

(11)   Line-657: please remove one “that” à OK

(12)   Line-706: “have already had contact” should be changed into have already been contacted à OK

Response 4: All corrections were made, except for number 8 which we couldn't find.

Point 5: Are the methods adequately described? Can be improved.

Response 5:

à Item 2.1 and 2.2, materials and methods, were summarized.

à A figure in topic 2.1 and a table in topic 2.6 were inserted to facilitate the understanding of the text.

Reviewer 2 Report

This is an interesting study evaluating the immunomodulatory effect of TNF antagonists and/or synthetic drugs, upon Toxotoxoplasmosis reactivation. Peripheral blood mononuclear cells (PBMC) isolated from autoimmune rheumatic diseases affected patient is under TNF inhibitors therapy, were evaluated after stimulation with antigens of Toxoplasma gondii, with anti-CD3/anti-CD28 or without stimulus at 48 and 96 hours. CD69+, CD28+ and PD-1 markers were evaluated alongside the intracellular expression of IFN-γ, IL-17, and IL-10 by CD4+ and cytokines IL-2, IL-4, IL-6, IL-10, IFN-γ, TNF-α, and IL-17. Main results indicate that the use of immunobiologicals alone did not promote the predisposing to toxoplasmosis reactivation.

1.    Despite being interesting, the work is difficult to read, as being too verbose in several sections. A large variety of notions, especially in the methods and discussion/conclusions, can be shortened in order to improve the quality of the work
2.    Numerous of typo error should be checked, e.g., line 644,  725 etc….
3.    Abstract, if not required, the numbered list of the abstract sections should be avoided
4.    The methods section is difficult to read as being too long.  It should be shortened in order to improve the readability of the work
5.    Authors should check the entire ms for the reference style, e.g., line 311
6.    The quality of all figures should be improved. For instance, figure 2 is quite difficult to understand
7.    The risk of reactivation of pathogenic infections, such as viral infections, has been reported in the context of TNF inhibitors therapy of autoimmune diseases, such a traumatic arthritis, and Merkel cell polyomavirus reactivation (PMID: 28174236). This important notion/work should be mentioned. Additional cases of viral reactivation during immunomodulatory therapy with TNF inhibitors has been reported for hepatitis B (PMID: 32897226, PMID: 19797507) and other latent viruses (PMID: 20142812)
8.    Lines 85-96 supporting references should be included. Similarly supporting references should be included in the methods section, including statistics. 
9.    Line 644 IL-13m?
10.    Line 668 “Another study..” reference
11.    Line 678 DMARD as well as other acronyms should be mentioned with their complete name when mentioned for the first itme

Author Response

Response to Reviewer 2 Comments

Point 1: Despite being interesting, the work is difficult to read, as being too verbose in several sections. A large variety of notions, especially in the methods and discussion/conclusions, can be shortened in order to improve the quality of the work

Response 1:

à The manuscript underwent an English review (the certificate of review will be attached on the journal page).

à Several excerpts were revised and summarized.

Point 2: Numerous of typo error should be checked, e.g., line 644,  725 etc….

Response 2: We did not find these specific typos, but we have proofread the text and hope that there are no other errors.

Point 3: Abstract, if not required, the numbered list of the abstract sections should be avoided.  

 Response 3: The Microsoft Word Biomedicines template file, provided by the journal in the "Instructions to authors" section was used. In this model, the summary had line numbers.

Point 4: The methods section is difficult to read as being too long.  It should be shortened in order to improve the readability of the work.

Response 4:

à Item 2.1 and 2.2, materials and methods, were summarized.

à A figure in topic 2.1 and a table in topic 2.6 were inserted to facilitate the readability of the text.

Point 5: Authors should check the entire ms for the reference style, e.g., line 311

Response 5: Corrections have been made.

Point 6: The quality of all figures should be improved. For instance, figure 2 is quite difficult to understand.

Response 6: In the text the figure actually looks a little small, however the resolution of the figure is at 600 dpi, as requested in the "Instructions for Authors" (300 dpi or higher). Separate files have been sent and will allow you to view the figures in higher quality.

Point 7: The risk of reactivation of pathogenic infections, such as viral infections, has been reported in the context of TNF inhibitors therapy of autoimmune diseases, such a traumatic arthritis, and Merkel cell polyomavirus reactivation (PMID: 28174236). This important notion/work should be mentioned. Additional cases of viral reactivation during immunomodulatory therapy with TNF inhibitors has been reported for hepatitis B (PMID: 32897226, PMID: 19797507) and other latent viruses (PMID: 20142812).

Response 7: References to Merkel cell polyomavirus risk (PMID: 28174236) and hepatitis B reactivation were included in the introduction (PMID: 19797507).

Point 8: Lines 85-96 supporting references should be included. Similarly supporting references should be included in the methods section, including statistics.

Response 8:

à New references on medical management in the treatment of rheumatic diseases were included, as well as the initial part was rewritten.

à In the methods, all protocols that were used or adapted and used from other authors were cited, such as reference 23 (Ribeiro et al, 2009), 24 (Bradford, 1976) and 25 (Hedman et al, 1993). The other protocols were followed from the manufacturers' recommendations when commercial kits were used, or were standardized in the immunology laboratory from Federal University of Triangulo Mineiro.

à New references were included in the statistical analyzes to explain the choice of normality test type.

Point 9:  Line 644 IL-13m?

Response 9: Adjusted.

Point 10:  Line 668 “Another study..” reference

Response 10: Adjusted.

Point 11:  Line 678 DMARD as well as other acronyms should be mentioned with their complete name when mentioned for the first itme.

Response 11:

à The term DMARDs was introduced in full in line 136, paragraph 4 of sub-item 2.1 of the Materials and Methods.

à Others terms were corrected, such as: avidity index (AI).

à line 224, item 2.5, PBMCs was simplified only with the acronym, as it had already been presented before.

à line 227, item 2.5, 4-(2-hydroxyethyl)-1-piperazineethanesulfonic acid (HEPES) buffer (Gibco).

Point 12:  Does the introduction provide sufficient background and include all relevant references? Can be improved.

Response 12: The introduction has been improved and references to Merkel cell polyomavirus risk (PMID: 28174236) and hepatitis B reactivation were included in the introduction (PMID: 19797507).

Point 13:  Are all the cited references relevant to the research? Must be improved.

Response 13: References to Merkel cell polyomavirus risk (PMID: 28174236) and hepatitis B reactivation were included in the introduction (PMID: 19797507).

Point 14:  Are the methods adequately described? Must be improved

Response 14:

à Item 2.1 and 2.2, materials and methods, were summarized.

à A figure in topic 2.1 and a table in topic 2.6 were inserted to facilitate the readability of the text.

Point 15:  Are the results clearly presented? Must be improved

Response 15:  We summarize part. of the results. However, as the study covers two types of analyses, between groups and between stimuli in the same group, we think it is important to emphasize the type of result mentioned.

Point 16: Are the conclusions supported by the results? Must be improved

Response 16: The conclusion of the study was rewritten.

Reviewer 3 Report

The manuscript by Cristhianne Molinero Ratkevicius Andrade et al., demonstrated an analysis of cellular and humoral immune response to Toxoplasma gondii in clinic samples with or without TNF blockers treatment. It tried to ask an important question of whether TNF blocker will lead to an increased risk of toxoplasmosis reactivation and whether some low-cost assays can help stratify high-risk patients for personal treatment plan. By testing samples treated with TNF blockers or not, the author found that treatment with TNF blockers doesn’t affect cellular and humoral immune response to Toxoplasma gondii in the assays they used.

Major issues:

1.     The author should include samples with toxoplasmosis reactivation to demonstrate their in vitro assays are robust.

2.     The readability of the manuscript is low and should be improved from at least three aspects.

a.     Avoid using very long sentences which make it difficult to understand. For example, in line 332-335, the sentence span 4 lines. The author should consider split it to multiple simple sentences. Also, some words can be removed to make information clearer. For example, ‘When the analysis between groups was performed’ (line 394) is totally unnecessary.

b.     Using words professionally. For example, in flow cytometry, we usually use ‘stain’ or ‘probe’ rather than ‘mark’ (Line 249 and Line 251). To ‘Verify’ (line 250) means to ‘justify’, but here I think the author try to say ‘monitor’. The authors used ‘in relation to’ when they tried to compare something, which I think is a miss-used phrase here.

c.     Adding more diagrams. The description in ‘2.6. Flow Cytometry’ is very confusing. Please add diagrams to showing how flow is done. Also, in the patient selection part, please add a diagram to showing the groups of ‘CG’, ‘SD’, and ‘IB’.

Author Response

Response to Reviewer 3 Comments

Point 1: The author should include samples with toxoplasmosis reactivation to demonstrate their in vitro assays are robust.

Response 1: We hypothesized that TNF blockers could jeopardize toxoplasmosis reactivation in order to raise awareness of the importance of monitoring patients under these conditions.

The Brazilian protocol does not propose the monitoring of toxoplasmosis in patients with autoimmune diseases using DMARDs, such as those studied in this manuscript. Thus, patients were not monitored for the risk of toxoplasmosis reactivation, nor were there any cases of toxoplasmosis reactivation in the hospital where we performed the study, during the patient recruitment phase.

Point 2: The readability of the manuscript is low and should be improved from at least three aspects.

  1. Avoid using very long sentences which make it difficult to understand. For example, in line 332-335, the sentence span 4 lines. The author should consider split it to multiple simple sentences. Also, some words can be removed to make information clearer. For example, ‘When the analysis between groups was performed’ (line 394) is totally unnecessary.

Response 2a: The manuscript underwent an English review (the certificate of review will be attached on the journal page).

à Several excerpts were revised and summarized.

  1. Using words professionally. For example, in flow cytometry, we usually use ‘stain’ or ‘probe’ rather than ‘mark’ (Line 249 and Line 251). To ‘Verify’ (line 250) means to ‘justify’, but here I think the author try to say ‘monitor’. The authors used ‘in relation to’ when they tried to compare something, which I think is a miss-used phrase here.

Response 2b: All mentioned inadequacies have been corrected.

  1. Adding more diagrams. The description in ‘2.6. Flow Cytometry’ is very confusing. Please add diagrams to showing how flow is done. Also, in the patient selection part, please add a diagram to showing the groups of ‘CG’, ‘SD’, and ‘IB’.

Response 2c: One figure was added: figure 1, in item 2.1, presents the division of groups and criteria for classification in each group. A table has also been added showing the antibodies used in flow cytometry.

Point 3:  Does the introduction provide sufficient background and include all relevant references? Can be improved.

Response 3: References to Merkel cell polyomavirus risk (PMID: 28174236) and hepatitis B reactivation were included in the introduction (PMID: 19797507).

Point 4:  Are all the cited references relevant to the research? Can be improved.

Response 4: References to Merkel cell polyomavirus risk (PMID: 28174236) and hepatitis B reactivation were included in the introduction (PMID: 19797507).

Point 5:  Are the methods adequately described? Can be improved.

Response 5:

à Item 2.1 and 2.2, materials and methods, were summarized.

à A figure in topic 2.1 and a table in topic 2.6 were inserted to facilitate the readability of the text.

Point 6:  Are the results clearly presented? Must be improved.

Response 6: We summarize part. of the results. However, as the study covers two types of analyses, between groups and between stimuli in the same group, we think it is important to emphasize the type of result mentioned.

Point 7: Are the conclusions supported by the results? Can be improved.

Response 7: The conclusion of the study was rewritten.

Round 2

Reviewer 1 Report

No thurther comments.

Reviewer 2 Report

The study can be published

Reviewer 3 Report

The manuscript is improved in this version and fully address my comments.